

# Concurrent Calculation of Radiative Transfer in the Atmospheric Simulation in ECHAM-6.3.05p2

Mohammad Reza Heidari[1], Zhaoyang Song[2], Enrico Degregori[3], Jörg Behrens[3], and Hendryk Bockelmann[3]

[1]Department of Informatics, Universität Hamburg, Hamburg, Germany
[2]School of Atmospheric Sciences, Sun Yat-sen University, and Key Laboratory of Tropical Atmosphere–Ocean System, Ministry of Education, Zhuhai, China
[3]German Climate Computing Center (DKRZ), Hamburg, Germany

**Correspondence:**
1. Mohammad Reza Heidari (heidari@informatik.uni-hamburg.de)
2. Zhaoyang Song (songzhaoy@mail.sysu.edu.cn)

**Abstract.** The scalability of the atmospheric model ECHAM6 at low resolution, as used in palaeoclimate simulations, suffers from the limited number of grid points. As a consequence, the potential of current high performance computing architectures cannot be used at full scale for such experiments, particularly within the available domain-decomposition approach. Radiation calculations are a relatively expensive part of the atmospheric simulations taking approximately up to over 50% of the total

runtime. This current level of cost is achieved by calculating the radiative transfer only once in every two simulation hours. In response, we propose to extend the available concurrency within the model further by running the radiation component in parallel with other atmospheric processes to improve scalability and performance. This paper introduces the *concurrent radiation scheme* in ECHAM6 and presents a thorough analysis of its impact on the performance of the model. It also evaluates the scientific results from such simulations. Our experiments show that ECHAM6 can achieve a speedup over 1.9x using

the concurrent radiation scheme. This empirical study serves as a successful example that can stimulate research on other concurrent components in atmospheric modeling whenever scalability becomes challenging.

# 1  Introduction

Earth system modeling has traditionally been a computationally demanding domain with a continual increase in complexity and resolution. It is a major application of high performance computing and represents a large variety of climate processes with

diverse computational profiles and disparate performance optimization requirements. A primary subsystem of a typical Earth System Model (ESM) is the atmospheric simulation, which resolves several physical processes including radiative transfer.

Radiative transfer is one of the most expensive parts in atmospheric simulations. This process is resolved to respond to the changing state of the chemical species which interact with the radiation (Balaji et al., 2016; Salby, 1996; Wallace and Hobbs, 2006). Solar energy is the driving force for the atmosphere through the radiative transfer, which is the only physical process

that is capable of exchanging energy between a planet like the Earth and the rest of the universe (Wallace and Hobbs, 2006).





Energy transfer in the atmosphere involves electromagnetic radiation in two widely separated wavelengths: *shortwave*, emitted by the sun, and *longwave*, emitted by the earth's surface and the atmosphere (Wallace and Hobbs, 2006; Salby, 1996). There are several atmospheric processes - including greenhouse gases, aerosols and clouds - that interact with electromagnetic radiation through the mechanisms of absorption, scattering and emission. The level of interaction strongly depends on the state of the
atmospherics particles (evolving by advection, cloud processes and chemistry) and the optical properties (the wavelengths and intensity) of the incident radiation (Wallace and Hobbs, 2006).

In principle, the absorption of solar radiation by the atmosphere and the earth's surface must be balanced by the longwave emission to space from the terrestrial radiation (Salby, 1996). It is crucial for atmospheric models to accurately represent the radiative transfer process (Rasp, 2019). Solving the problem is in essence straightforward (Wallace and Hobbs, 2006).
However, this can be quite computationally demanding in practice, despite the simplifying approximations adopted in the radiation component (Balaji et al., 2016; Wallace and Hobbs, 2006). As a result, in most climate models around the world, this component is not called in every time step (Balaji et al., 2016). On the contrary, it is calculated at a coarser time step than the rest of atmospheric physics, entirely pursuing a performance improvement rather than fulfilling any other technical objective (Balaji et al., 2016).

Over the years, various techniques have been used to represent radiative transfer in different models and maintain its calculation cost within acceptable limits. Morcrette (2000) discusses the effect of the temporal or spatial sampling techniques of the radiation inputs on the forecasts and analyses at the European Centre for Medium-Range Weather Forecasts (ECMWF). In a different approach, the radiative calculations are computed on a coarser grid than the one on which all other physical processes are implemented. Morcrette et al. (2008) report the implementation of a reduced grid for radiation in the ECMWF Integrated
Forecasting System (IFS) model.

Another proposal is based on the reorganization of the radiation component within the atmospheric models. In this approach, the radiative transfer is calculated in separate tasks in parallel with the rest of the model, in pursuit of improved scalability and performance. In the classical atmospheric modeling, all physical processes including solar radiation are resolved sequentially with respect to each other; thus creating a prohibitively long latency in the overall runtime. Reorganizing the radiation com-
ponent as separate parallel tasks allows for the simultaneous calculation of radiative transfer along with the other atmospheric processes and removes the long response latency from the component. Mozdzynski and Morcrette (2014) reports a reorganization of the radiation calculations in IFS at ECMWF and demonstrates the *radiation-in-parallel* configuration, in which calculating radiative transfer is performed on separate MPI (Message Passing Interface) processes in parallel with the rest of the model.

In a similar effort but in a broader sense, Balaji et al. (2016) proposes the *Coarse-grained Component Concurrency* (CCC) to increase the level of concurrency within ESMs. This approach suggests reorganizing more lower- and higher-level components in parallel with each other and exploiting fine-grained parallelism within each component individually. Additionally, the report demonstrates the result of applying this approach to the radiation component of the Geophysical Fluid Dynamics Laboratory (GFDL) Flexible Modeling System (Balaji, 2004). In this use case, the atmospheric radiative transfer is configured to run in





parallel with the atmospheric dynamics and all other atmospheric physics. This technique uses shared-memory parallelism and divides the available threads in an OpenMP (Open Multi-Processing) region between the parallel components.

This paper concentrates on the performance optimization of the radiation scheme in ECHAM6, which is the sixth generation of the atmospheric general circulation ECHAM (Stevens et al., 2013). The model was developed at the Max Plank Institute for Meteorology (MPI-M) in Hamburg. It is the traditional atmospheric component of the coupled Earth System Model MPI-

ESM, as indicated in Figure 1 and described by Giorgetta et al. (2013). ECHAM6 benefits from spectral and finite difference methods in five different grid resolutions, ranging from the coarse (CR) and low resolution (LR) to the very high resolution (XR). The CR or T31 is a truncation to 31 wave numbers in the spectral part and corresponds to a horizontal spatial resolution of 96×48 points in longitude and latitude. The LR or T63, however, corresponds to 192×96 points. Its special prominence in this research is due to its application in the German climate modeling initiative (PalMod), which aims at simulation of

a complete glacial cycle (i.e., about 120000 years) from the last interglacial to the Anthropocene (https://www.palmod.de). There, however, remains a serious caveat as to the feasibility of such an ambitious project which should be acknowledged in advance. In particular, a major concern has been raised over the poor performance of ECHAM6 suffering from the limited number of grid points at setups used in palaeoclimate simulation. For this reason, the performance optimization of the model is instrumental in ensuring the viability of such long-time simulations.

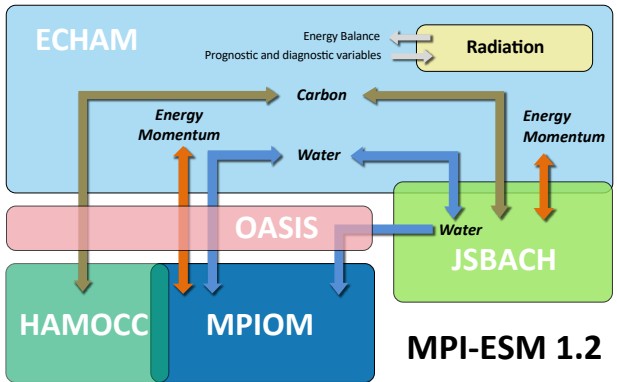

**Figure 1.** The coupling architecture of the Earth system model MPI-ESM

Experiments reveal that the radiation component is one of the most expensive computational parts in ECHAM6, at least for palaeoclimate simulation. Endeavors to adopt higher optimized radiative calculations will, therefore, be opportune for PalMod experiments. In response, two solutions have been investigated in parallel within the PalMod project to alleviate the computational burden of the radiative transfer on the atmospheric simulations in ECHAM6: single-precision arithmetic and *concurrent radiation scheme*. The technique of mixed-precision calculations is a practice of reducing time-to-solution

of scientific algorithms whenever lower accuracy is permitted. Empirical studies have revealed that radiation calculations



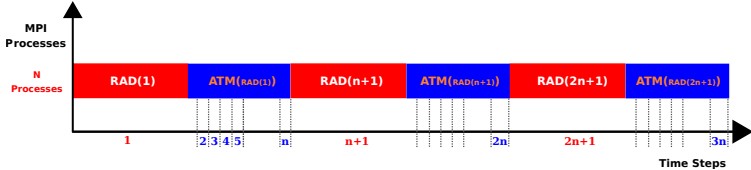

**Figure 2.** The organization of the classical radiation scheme in ECHAM6: the radiative transfer is resolved sequentially with respect to the other atmospheric physics and dynamics and it is stepped forward at a slower rate than the other atmospheric processes.

can benefit from reduced-precision arithmetic. Cotronei and Slawig (2020) discuss the results of applying single-precision arithmetic to the radiation calculations in ECHAM6. They indicate that this mathematical treatment accelerates the component by about 40% while the overall runtime of the model reduces by 18%.

This paper, on the other hand, presents a report on the concurrent radiation scheme applied to the atmospheric model
ECHAM6 and performs a thorough analysis, in light of the new scheme, on the performance and stability of the model. In contrast to the OpenMP approach used by Balaji et al. (2016), the concurrent radiation scheme opts for the MPI parallelization in order to fully exploit the potential of higher concurrency in the model. In addition, encapsulating the radiation calculations in a distinct component and name space realizes the idea of separation of concerns (SoC), which results in more degrees of freedom. An immediate benefit includes the independent development and optimization of the radiation component from the
main model pursuing higher throughput - which is essential for the ambitious long simulation runs of the PalMod project. This architectural merit also enables the potential of combining the virtues of the concurrent radiation scheme with other appropriate optimized solutions such as "single-precision arithmetic in ECHAM radiation" (Cotronei and Slawig, 2020) in the future.

The paper is organized as follows: Section 2 describes the classical approach to the radiation calculations in ECHAM6. In Section 3, we introduce the solution of a new radiation scheme in the model. Finally, the performance analysis and the scientific
evaluation of the new scheme are presented in section 4 and 5, respectively.

## 2  The Classical Radiation Scheme in ECHAM6

Radiative transfer in ECHAM6 is represented with PSrad/RRTMG (a postscript to the Rapid Radiative Transfer Model for GCMs (Pincus and Stevens, 2013)) for both shortwave and longwave parts of the electromagnetic spectrum (Stevens et al., 2013). The radiation component is one of the most expensive components within atmospheric physics. As a result, this com-
ponent is stepped forward at a slower rate than the rest of the atmospheric physics in ECAHM6 as well as most of the climate models around the world (Balaji et al., 2016). In its flagship configuration, ECHAM6 updates the optical properties of radiation every two hours, except for very high-resolution (T255) simulations where the radiation is calculated hourly. Figure 2 shows the organization of the classical radiation scheme in ECHAM6. As it is apparent in this figure, the radiative transfer is generally calculated once in every n normal atmospheric time steps, i.e. $\Delta t_{\mathrm{rad}} = n * \Delta t_{\mathrm{atm}}$.

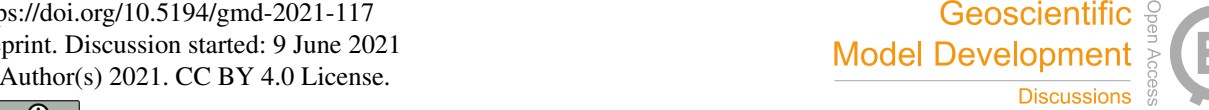

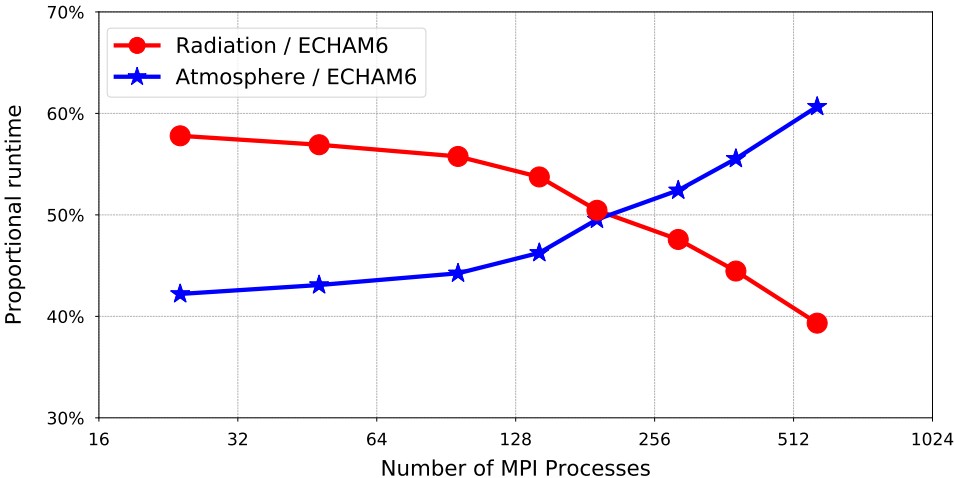

**Figure 3.** The relative time contribution of radiation calculations in ECHAM6 (using the classical radiation scheme) in the CR simulations.

In this scheme, the radiation results are used beyond the state of the input tracers, which may be as much as $\Delta t_{\mathrm{rad}} - \Delta t_{\mathrm{atm}}$ (approximately two hours) behind. Figure 2 schematically gives a clear account of what happens. As it shows, there are multiple normal atmospheric time steps between two consecutive radiation time steps. The atmospheric calculations are provided with old feedback from the radiation component within the normal time steps. On non-radiation time steps in between an update of the optical properties, longwave irradiance is rescaled based on the surface temperature while shortwave irradiance is rescaled

by the zenith angle (Stevens et al., 2013). Infrequent calculations of the radiative heating may result in numerical instability in climate models, as described by Pauluis and Emanuel (2004). As pointed out by Balaji et al. (2016), the use of the lagged state can be viewed as a potential source of discrepancy between the cloud field and the "cloud shadow field" seen by the radiation component. This, therefore, introduces numerical errors in atmospheric models and becomes considerably worse at higher resolutions (Xu and Randall, 1995; Balaji et al., 2016). Although the choice of larger radiation time steps ($\Delta t_{\mathrm{rad}} >$

$\Delta t_{\mathrm{atm}}$) evidently reduces the overall runtime of simulations, the radiative portion is yet considered relatively high for some configurations in ECHAM6. As shown in Figure 3, the radiative calculations take up from almost 40% to 58% of the total simulation time at the CR resolution (which is also one of the PalMOD settings), depending on the number of MPI processes assigned to the model.

    The entire crux of the problem in the classical radiation scheme can be attributed to the sequential organization of the compo-

nents inside the model. In this architecture, atmospheric processes are stepped forward one at a time in every time step and the computation time of each component directly contributes to the overall simulation time. In particular, the radiation component significantly delays the following calculation of other atmospheric physics and dynamics during the entire course of simulation. As it is apparent in Figure 2, this architecture prolongs the radiation time step in proportion to the high computational cost of





the radiation component. It will be shown in the next section that this long response time of the radiation calculations is not,
however, inevitable and can be avoided by reorganizing the component inside the model.

Moreover, the sequential organization of components in the classical ECHAM6 creates another obstacle that hinders the op-
timization of the model. In fact, ECHAM6 traditionally benefits from MPI and implements domain decomposition parallelism
to expedite the computations. As Figure 4 shows, the radiation calculations display a higher scalability than the main model
in this framework. This can be attributed to the columnwise organization of the atmospheric physics and the embarrassingly
parallel nature of the workload. In other words, since the individual columns (iterating the k index in an (i, j, k) discretiza-
tion) have no cross-dependency in (i, j), it allows for fine-grained parallelism (Balaji et al., 2016). This is the reason why
the radiation component can intrinsically scale better beyond the limitations of the main model. Such a higher scalability is
instrumental in reducing the high computational cost of the radiation calculations. However, the sequential architecture of the
classical ECHAM6 restrains the benefit by forcing the radiation component to use the same computational resources as the rest
of the model. As a consequence, the component is hindered by the limited scalability of the whole model.

In the next section, however, it will be shown that re-organization of the components in ECHAM6 is essential to improve
the overall performance of the model.

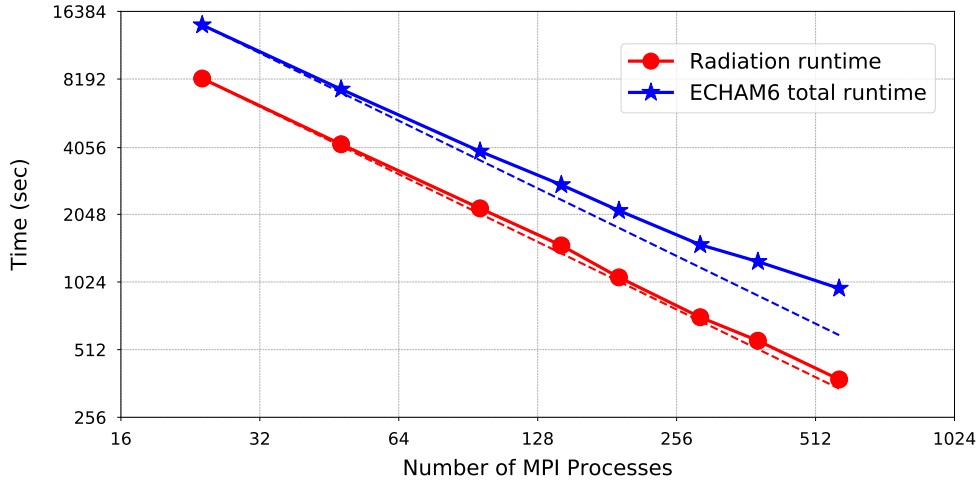

**Figure 4.** The scaling curve of the radiation component vs. the atmospheric model ECHAM6 shows that the radiative transfer has a higher
scalability and keeps scaling beyond the limitations of the main model.

# 3 The Concurrent Radiation Scheme

It was discussed in the previous section that, even in light of larger radiation time steps, radiation calculations yet impose a
daunting cost on the atmospheric simulation at coarse and low resolution in ECHAM6. It was also shown that the sequential





treatment of resolving atmospheric processes is implicated in the long response time of the radiation component in every radiation time step and restricts the benefit of the higher scalability of the radiation calculations.

The concurrent radiation scheme, on the other hand, puts forward a feasible solution to the problem. As shown in Figure 5, it implements an additional level of parallelism inside the model by applying coarse-grained component concurrency to the radiation calculations. This approach eliminates the high response latency from the radiation time step and paves the way for a higher scalable model. In contrast to the classical scheme, the concurrent radiation scheme starts resolving the radiative transfer much earlier before the next radiation time step arrives. As a result, the main model receives feedback from the radiation component much faster upon the request. This technique minimizes the response latency of the component and reduces the overall simulation time. In this approach, the radiative transfer is calculated concurrently with other atmospheric processes along the course of normal time steps. The coupling fields are also exchanged between the radiation component and the main model within the radiation time steps but without the typical delay experienced in the classical scheme. Figure 6 describes a method for casting the radiative transfer as a concurrent component using distributed memory computing. This technique organizes the radiation component and the main model on separate MPI processes and enables the concurrent calculation of the radiative transfer and other atmospheric processes.

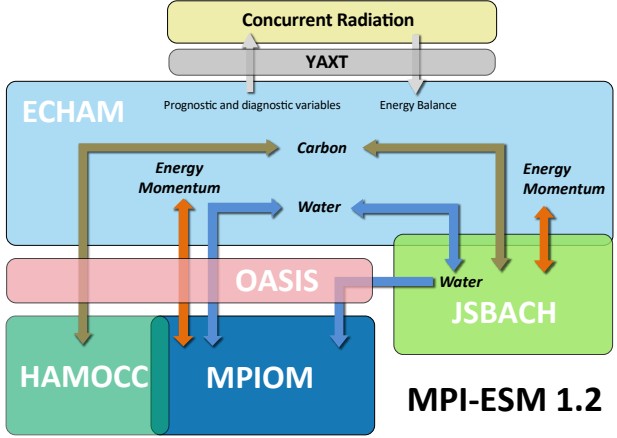

**Figure 5.** The concurrent radiation scheme adds a new level of parallelism inside the atmospheric model ECHAM6 and increases coarse-grained component concurrency in the coupling architecture of the Earth system model MPI-ESM.

Due to the close time-dependency of radiation on evolving model fields, the concurrency between the atmosphere and radiation component is only enacted between consecutive radiation time steps. In addition, the synchronization between the concurrent components takes place during each radiation time step. As it is shown in the next section, the interprocess synchronization overhead is, however, negligible compared to the cost of the radiation calculations. The data exchange between the radiation and the atmospheric model benefits from the communication library YAXT (Behrens et al., 2014), which was developed at German Climate Computing Center (DKRZ) in Hamburg. YAXT simplifies the formulation of the communica-

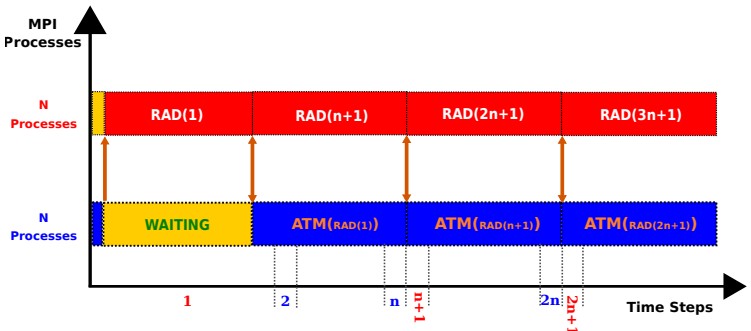

**Figure 6.** The re-organization of the radiation component in parallel with the rest of atmospheric physics and dynamics in ECHAM6. In the first (radiation) time step, ATM (the main model) sends the input data to RAD (the radiation component) but it has to wait long until it receives the results from RAD. In the following radiation time steps, however, the data exchange takes place immediately one after the other (i.e. ATM first receives the results of radiation calculations and immediately provides the input data to RAD for the next radiation calculations.). This way, ATM is supposed to experience a mimimum idle time when it interacts with RAD.

tion problem and generates suitable communication objects to efficiently execute the data exchange. The library is specially suited for bulk communication as given in this use case. This is due to the automatic generation of MPI datatypes that enable direct access to model data without requiring additional data copies or pack/unpack overhead to create messages. YAXT is built on top of MPI and takes high level descriptions of arbitrary domain decomposition and automatically derives an efficient
160    collective data exchange. Figure 7 shows the use of the YAXT library for coupling two concurrent components with different domain decomposition.

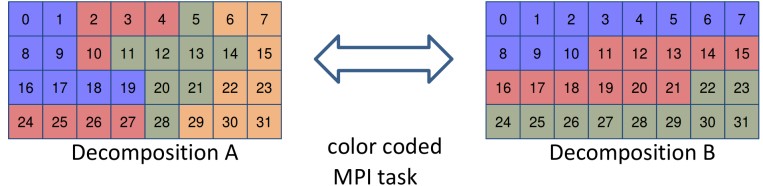

**Figure 7.** YAXT library facilitates MPI communication between concurrent components with different domain decomposition layouts.

In addition, running the concurrent components on separate MPI processes is a prerequisite to applying arbitrary domain decomposition to the radiation calculations. It offers a way forward to scale the component beyond the limits traditionally imposed by the atmospheric model. Hence, the concurrent radiation scheme prepares the ground for improving the physical
165    consistency between the radiative and physicochemical atmospheric states. This approach can accordingly minimize the discrepancy between $\Delta t_{\mathrm{rad}}$ and $\Delta t_{\mathrm{atm}}$ (Pauluis and Emanuel, 2004; Xu and Randall, 1995; Balaji et al., 2016). Furthermore, the independent resource allocation feature is essential for an efficient load balancing between concurrent components and parallel efficiency of the model. The next section presents these merits in more detail with some concrete examples.



# 4   Results I: Performance analysis

This section presents the performance evaluation of the concurrent radiation scheme in ECHAM6 in comparison with the classical approach. It should be emphasized that the new scheme utilizes almost the same original implementation of radiation calculations with a radically different orchestration. The new organization, therefore, exerts a major impact on the overall simulation time rather than the pure computational performance of the radiation component. In consequence, the performance evaluation presented in this section explicitly aims at the assessment of the whole model and will not be limited to the radiation component. For the purpose of this study, a new version of ECHAM6 (based on ECHAM-6.3.05p2) is deployed with both classical and concurrent radiation schemes which can be configured to calculate the radiative transfer with or without separate MPI processes. Experiments are performed on the *Mistral* supercomputer at DKRZ on a machine configuration with Intel Haswell processors (E5-2680v3 12C 2.5GHz) and Infiniband high speed interconnect. All runs allocate a layout of one MPI process per CPU core on the computing nodes equipped with two processors which are exclusively dedicated to the experiments. The performance results presented in this paper are obtained from the Atmospheric Model Intercomparison Project (AMIP) experiments performing simulations at the CR from 1976 to 1981.

To understand the impact of the concurrent radiation scheme on the overall performance, it is useful to extract the scaling curves of the model with both radiation schemes and study the gained speedup via the concurrent radiation scheme. For this purpose, the frequency at which the radiative transfer is calculated is by default every two hours in all runs, i.e. $\Delta t_{\mathrm{rad}} = 8 * \Delta t_{\mathrm{atm}}$ with the normal time step of 15 minutes. Additionally, once the model is configured to use the concurrent radiation scheme, an equal number of MPI processes, and thus identical domain decomposition, is assigned to the radiation component and the main model.

Figure 8 compares two scaling curves which reflect the performance of the model with the classical (the blue curve) and concurrent radiation schemes (the red curve). The horizontal axis shows the total number of MPI processes allocated by the model. It is worth emphasizing that ECHAM6 (using the classical radiation scheme) uses the same MPI processes to calculate radiative transfer and other atmospheric processes. However, when the model is configured to use the concurrent radiation scheme, half of the allocated MPI processes are exclusively dedicated to the radiation calculations and the other half to the rest of the atmospheric physics and dynamics. The Y-axis, on the other hand, reflects the throughput of the model in terms of the number of simulated years per day (SYPD). As it can be inferred from Figure 8, ECHAM6 can achieve only a maximum performance of 450 SYPD at 576 MPI processes using the classical radiation scheme at the CR resolution. However, it yields a significant improvement using the concurrent radiation scheme and reaches a maximum performance of 734 SYPD at 1152 MPI processes. It is noteworthy that, due to the limited number of grid points at the CR resolution, running the classical model at higher domain decomposition is not justified theoretically. Needless to say, it does not attain any significant performance improvement in practice either, as asserted in Figure 4 where the scaling curve of ECHAM6 tends to flatten towards the end. This should explain why the blue curve in Figure 8 stops at 576 MPI processes, as opposed to the red curve scaling beyond. On this account, the concurrent radiation scheme acquires a new significance as it becomes conducive to higher scalability of the model.



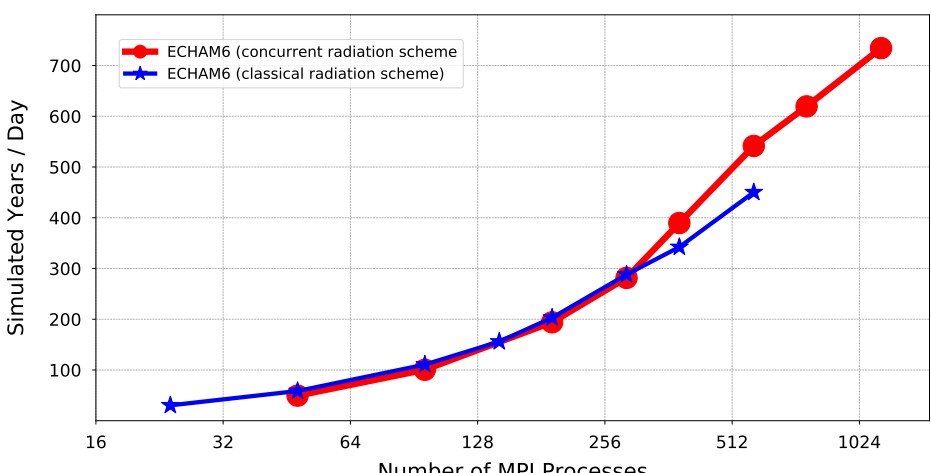

**Figure 8.** The scaling curves of ECHAM6 using the classical and concurrent radiation schemes.

Figure 9 displays the speedup of the model across the scaling curves (shown in Figure 8). The red curve indicates the actual speedup of the model, which is the ratio of the overall performance of the model when the concurrent radiation scheme is adopted to its classical performance. The X-axis indicates the total number of allocated MPI processes if the concurrent radiation scheme is used by the model. However, the model allocates half of the MPI processes shown at the X-axis when it adopts the the classical radiation scheme. The red curve shows that the model achieves an actual speedup ranging from a minimum 1.6x to over 1.9x when it benefits from the concurrent radiation scheme. The blue curve, however, shows the asymptotic speedup which the model would achieve if there were no cross-dependency, and thus no communication latency, between the radiation component and the main model.



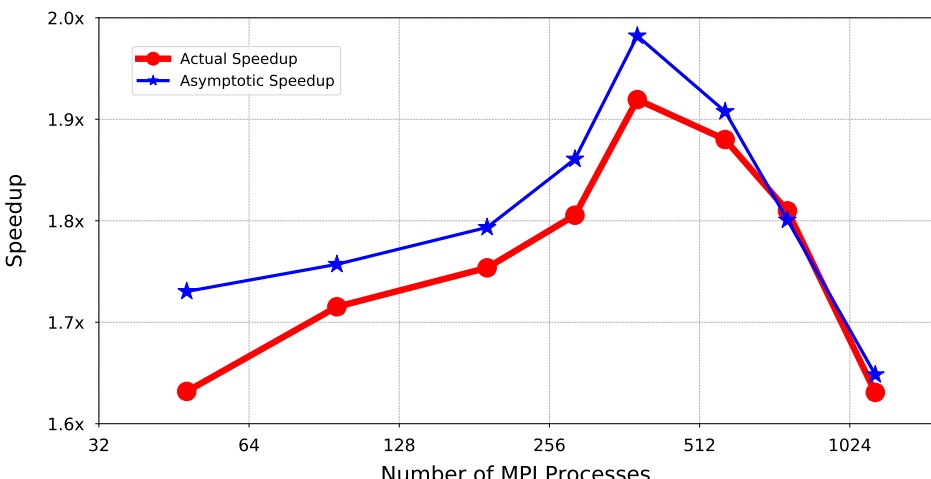

**Figure 9.** The overall speedup of ECHAM6 using the concurrent radiation scheme and comparing it with the asymptotic speedup.

Moreover, the coupling fields between the radiation component and the rest of the model are exchanged at every radiation time step and potentially contribute to the overall simulation time. There is nevertheless mounting evidence in Figure 10 indicating that the communication overhead compared to both radiation and total runtime of the model is negligible (less than 1.4%) and it has, therefore, little impact on the total simulation time. Consequently, the performance of the model (using concurrent radiation scheme) is mainly affected by the relative cost of the radiation calculations, which has a strong dependency on the number of allocated computing resources.



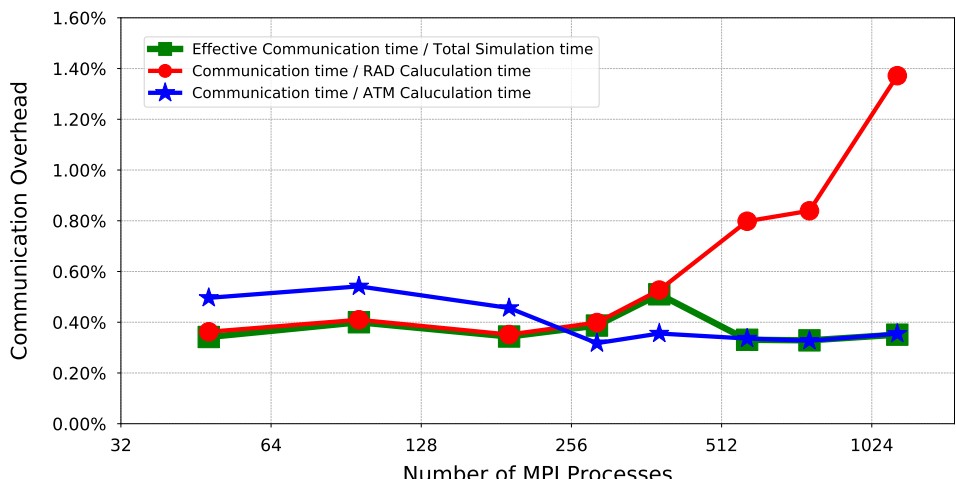

**Figure 10.** Comparing the communication costs of data exchange between the radiation component and the main model to the computation time of the radiative transfer (the red curve) and other atmospheric processes (the blue curve) and to the overall simulation time (the green curve).

Figure 11 shows the efficiency of the resource utilization when the model runs the concurrent radiation scheme. The parallel efficiency of the concurrent radiation scheme is defined as the ratio of the methodical speedup $S$ to the relative number of the allocated resources $R$, as shown below. Here, the methodical speedup means the improved runtime by making use of concurrency; in contrast to the classical definition of speedup where additional resources are used for the same computation.

$$\text{Methodical Speedup} = S = \frac{Ts}{Tp} = \frac{\text{(maximum simulated years per day with the concurrent radiation scheme)}}{\text{(maximum simulated years per day with the classical radiation scheme)}} \quad (1)$$

$$\text{Ratio of allocated resources} = R = \frac{\text{(resources assigned to the concurrent radiation scheme)}}{\text{(resources assigned to the classical radiation scheme)}} = \frac{2N}{N} = 2 \quad (2)$$

$$\text{Parallel Efficiency} = E = \frac{S}{R} = \frac{S}{2} \quad (3)$$

As visible in the figure, the model achieves a parallel efficiency of 80% or more across the scaling curve. However, attaining the maximum parallel efficiency requires an optimal distribution of workload among MPI processes. A close investigation reveals a load imbalance between the concurrent components inside the model. This problem starts appearing when the radiation and other atmospheric calculations are configured to use identical domain decomposition and allocate an equal number of MPI processes. The measured parallel efficiency in Figure 11 suggests that the MPI processes assigned to the radiation calculations experience an idle time during the course of simulation.

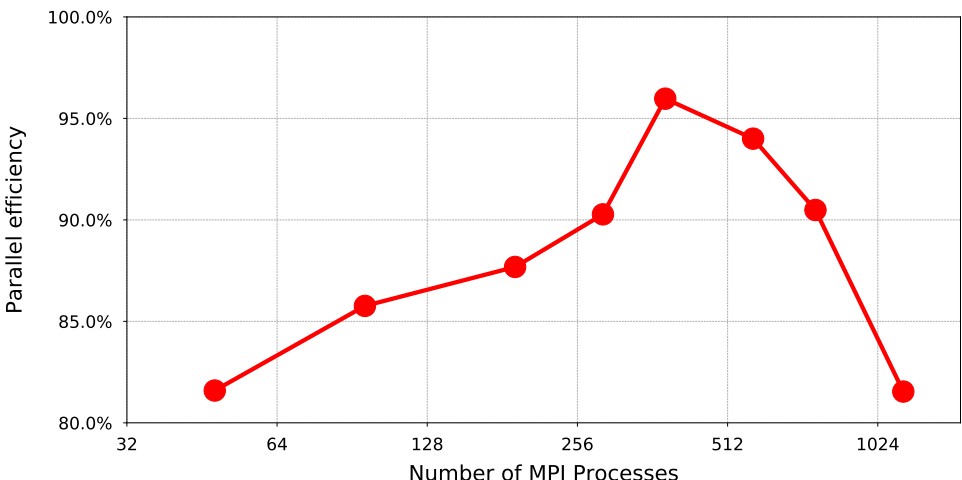

**Figure 11.** The parallel efficiency of ECHAM6 (using the concurrent radiation scheme).

Figure 12 depicts the length of time that the concurrent components have to remain idle, waiting for the slower component to catch up and become ready to exchange the radiation results. The red and blue curves show the respective idle times that the radiation component (denoted by RAD) and other atmospheric processes handled by the main model (denoted by ATM) experience. The idle times appear when the radiative transfer is resolved faster or slower than the other atmospheric calculations. At lower numbers of MPI processes, as shown in Figure 12, calculating the radiative transfer takes longer and it, thus, forces ATM to wait relatively long in each radiation time step for feedback from RAD. At higher numbers of MPI processes, however, RAD scales better and finishes the calculations faster than ATM. It therefore has to wait for the arrival of the next radiation time step so that the radiation results can be transferred to the main model. It should be noted that Figure 12 shows the maximum length of time that an MPI process (assigned to RAD or ATM) has to wait for its peer MPI process (assigned to the other component) until it catches up. From Figure 12, it can reasonably be inferred that the total idle time experienced by ATM and RAD becomes minimum at 384 MPI processes. This can also explain why the parallel efficiency in Figure 11 has an extremum at this point. It is also apparent in Figure 12 that the radiation component experiences a longer idle time as the the number of MPI processes increases. This behavior accounts for the higher scalability of the radiation component, which was already reflected in Figure 4.

The load imbalance between the concurrent processes is directly affected by the number of MPI processes assigned to the radiation component as well as the frequency at which the radiative transfer is calculated. Figure 13 schematically illustrates a contrived configuration in which the radiation component is forced to remain idle almost half of the total runtime. To remove such an idle time, Figure 14 provides an inspiring example. It takes advantage of the higher scalability of the radiation component and reduces the radiation time step to a half. This new configuration eliminates the idle time from the MPI processes assigned to the radiation component and hence removes the load imbalance between the concurrent components successfully.





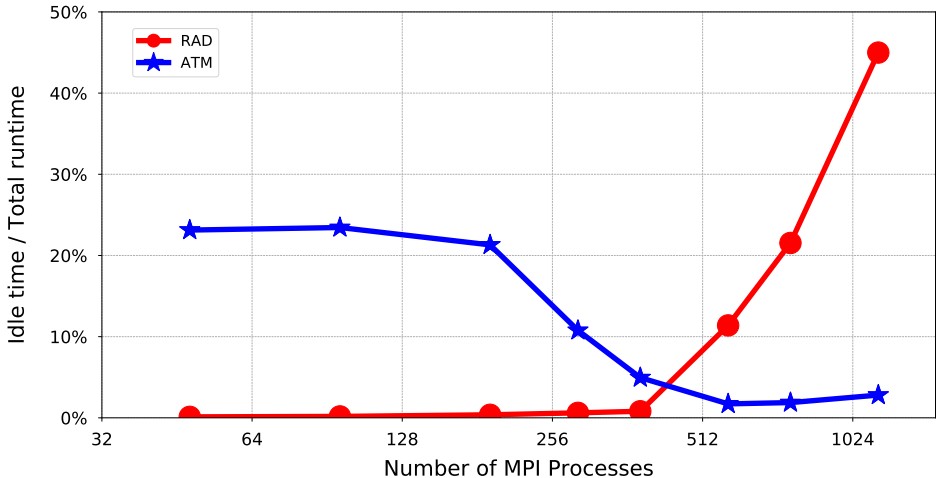

**Figure 12.** The red curve (RAD) shows the increasing idle time of the MPI processes responsible for resolving radiative transfer. It suggests that RAD is a dominant computation in all the configurations before 576 MPI processes and does not wait for ATM. The blue curve (ATM), on the other hand, shows that resolving other atmospheric physics and dynamics is not a dominant computation for all the configurations before 576 MPI processes and ATM therefore experiences a long idle time. However, ATM becomes dominant towards the end, inflicting a long idle time on RAD. Yet the idle time of ATM is not lifted completely. This can be attributed mainly to the unavoidable waiting time of ATM in the first (radiation) time step (as reflected in Figure 6) and some infrequent (slightly) longer radiation time steps.

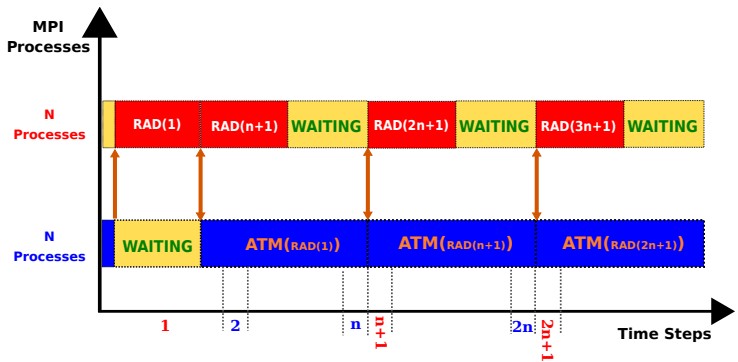

**Figure 13.** A contrived model configuration in which the same domain decomposition is assigned to the radiation calculations and the main model. Here, this setup imposes an idle time on the radiation component, leading to an inefficient resource usage.

As a consequence, the resource utilization of the model improves significantly and attains a parallel efficiency close to 100% without affecting the achieved speedup. This example presents a viable solution to decrease the gap between $\Delta t_{\mathrm{rad}}$ and $\Delta t_{\mathrm{atm}}$ in pursuit of a more consistent atmospheric model.



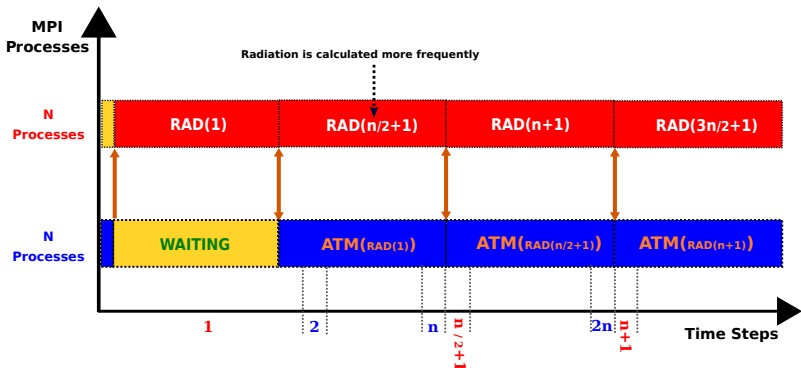

**Figure 14.** A model configuration for assigning identical domain decomposition to the radiation component and the main model but reducing the radiation time step to a half ($\Delta t_{\mathrm{rad}} = n/2 * \Delta t_{\mathrm{atm}}$), hence improving the parallel efficiency of the model.

In the general case, however, the concurrent radiation scheme puts forward an alternative solution for load balancing between the MPI processes inside the model. Figure 15 suggests a configuration in which the radiation calculations adopt coarser domain decomposition compared to the main model. In this setup, the radiation time step is kept unchanged but a smaller number of MPI processes are assigned to the radiation component. This arrangement achieves a more efficient resource utilization.

By the same token, Figure 16 proposes another useful model configuration which applies finer domain decomposition to the radiative transfer. It therefore enables a more frequent calculation of the radiative transfer in the quest for coupling the radiation component to the other atmospheric physics and dynamics at every normal time step (i.e. $\Delta t_{\mathrm{rad}} = \Delta t_{\mathrm{atm}}$). This proposal can ultimately bring the model to the physical consistency between the radiative and physicochemical atmospheric states by allocating a large number of computational resources.

It is notable that the current implementation of the concurrent radiation scheme in ECHAM6 already provides the technical support for the adoption of both configurations proposed in Figure 15 and Figure 16. In particular, the YAXT library simplifies the data exchange between the concurrent components with arbitrary domain decomposition. The scientific viability of these schemes, however, requires further investigations and the results will be presented in a follow-up paper.

# 5 Results II: Scientific evaluation

We evaluate the concurrent radiation scheme in ECHAM 6.3.05 version at the CR and LR configurations, hereafter termed as $\mathrm{CR_{CRR}}$ and $\mathrm{LR_{CRR}}$, which differ in their horizontal grid spacings. Both $\mathrm{CR_{CRR}}$ and $\mathrm{LR_{CRR}}$ share the vertical resolution of 47 levels. The classical radiation scheme has been tuned to optimize the simulated climate (Mauritsen et al., 2019). However, the concurrent radiation scheme at these two configurations have not been individually tuned. The parameters of the convection scheme, e.g., the convection conversion rate for cloud water to rain in the concurrent radiation scheme are the same as in the classical scheme. The evaluation and documentation of the concurrent radiation scheme in ECHAM6.3 is based on the Atmo-





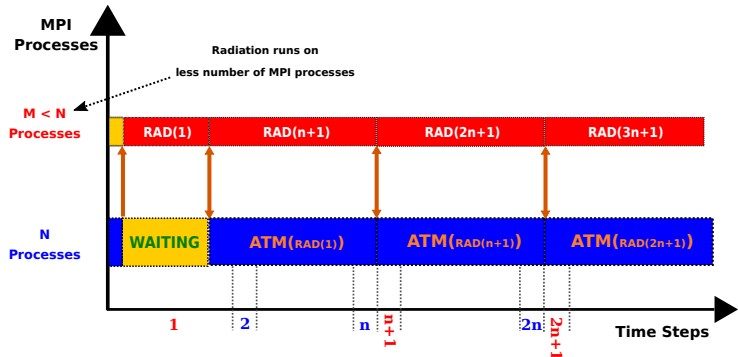

**Figure 15.** A model configuration for assigning coarser domain decomposition to the radiation calculations. This setup allows for allocating a lower number of MPI processes to the radiation component and hence improves the parallel efficiency of the model.

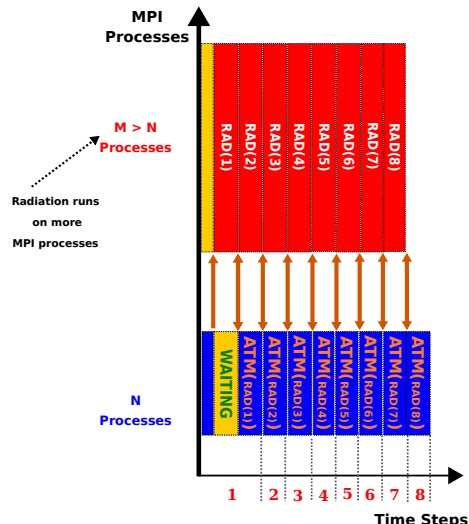

**Figure 16.** A model configuration for assigning finer domain decomposition to the radiation calculations. This setup allows for allocating a higher number of MPI processes to the radiation component and hence prepares the ground for closing the gap between $\Delta t_{\text{rad}}$ and $\Delta t_{\text{atm}}$ in order to create consistency between the radiative and physicochemical atmospheric states.

275  spheric Model Intercomparison Project (AMIP) historical experiments. Experiments were performed according to the AMIP protocol (Eyring et al., 2016). The historical forcings include the emissions of short-lived species and long-lived greenhouse gases, solar forcing, stratospheric aerosol forcing, monthly sea surface temperatures and sea ice concentrations. Experiments span from 1960 to 2013. The monthly output from 1980 to 2013 are taken for analysis. AMIP experiments performed with the classical ECHAM6.3.05, as well as the observations and reanalysis, serves as a reference. Observational and reanalysis data

280  used in this paper are listed in Table 1.





**Table 1.** Observation and reanalysis data for model evaluation.

| Variable | Validation data | Period | Reference |
|---|---|---|---|
| Surface land Temperature | HADCRU4 | 1980-2013 | Jones et al. (2012) |
| Total Precipitation | GPCP | 1980-2013 | Adler et al. (2003) |
| Surface LW radiation | CERES-EBAF | 2001-2013 | Loeb et al. (2012) |
| Surface SW radiation | CERES-EBAF | 2001-2013 | Loeb et al. (2012) |
| Zonal mean temperature | ERA-interim | 1980-2013 | Dee et al. (2011) |
| Zonal mean zonal wind | ERA-interim | 1980-2013 | Dee et al. (2011) |
| Air pressure at sea level | ERA-interim | 1980-2013 | Dee et al. (2011) |

## 5.1 Mean State

We first evaluate the simulated mean climate by the concurrent radiation scheme. CRUTEM4 observations are interpolated to the model grids for the comparison against simulated surface-air temperature (SAT). Across all configurations, global annual bias in SAT over land exhibits similar spatial patterns (Figure 17). General warm bias occurs over eastern Siberia, Central Asia, Tibetan Plateau, North America and central Australia, while sporadic cold biases exist in south Asia, Africa and northern South America. For a rough assessment, global root-mean-square-error (RMSE) is calculated for SAT. Horizontal resolution exhibits larger impact on the SAT bias than the radiation scheme. Both radiation scheme shows smaller SAT bias in LR relative to CR. The concurrent radiation scheme illustrates a somewhat ambiguous reduction in the annual SAT bias.

All experiments suffer from severe precipitation bias over the tropics as compared to the GPCP satellite measurements (Figure 18). Specifically, excessive precipitation is simulated over the tropical Indian Ocean, western tropical Pacific and western tropical Atlantic Ocean, while deficient precipitation occurs over the east tropical Pacific Ocean, South America and east tropical Atlantic Ocean. Consistent with the response of SAT bias, the global RMSE decreases along with a refinement of horizontal resolution, whereas the concurrent radiation scheme leads to larger bias in the precipitation. Particularly, the precipitation bias over the equatorial region of South America and western equatorial Africa increases in the concurrent radiation (Figure 18a and 18c), which is likely linked to the intensification of local SAT bias (Figure 17a and 17c). Clouds play a complex and crucial role in the earth's radiation budget, which significantly affects the simulated surface climate. To understand the biases in SAT, we evaluate the errors in cloud feedback by estimate cloud-radiative-forcing (CRF) following (Cess et al., 1990).



**a** CR$_{CRR}$ minus CRU (RMSE= 2.71)  **b** CR$_{SEQ}$ minus CRU (RMSE= 2.77)

**c** LR$_{CRR}$ minus CRU (RMSE= 2.17)  **d** LR$_{SEQ}$ minus CRU (RMSE= 2.29)

-6 -5 -4 -3 -2 -1 0 1 2 3 4 5 6  K

**Figure 17.** Annual bias in surface-air temperature (SAT) over land (K) in the (a, c) concurrent and (b, d) classical radiation experiments and Research Unit time series 4.01 data set (CRU TS; Harris et al., 2014) for the period 1980–2013.

$$CRF = F_{all-sky} - F_{clear-sky} \qquad (4)$$

where F$_{all-sky}$ refers to radiative flux at the surface and F$_{clear-sky}$ represents the radiative flux assuming the absence of clouds.
CRF is positive when clouds warm the surface atmosphere, and vice versa. The radiative effect of clouds consists of two competing components on the surface radiation budget: warming impact on the surface through the emission of longwave (LW) radiation and cooling impact by shading the shortwave (SW) radiation. The spatial distribution of total CRF biases over land is quite complex due to the variations in the orography (Figure 19). The magnitude of CRF biases over land, however, is generally modest, except over the Tibetan Plateau. In contrast, CRF biases over the ocean are relatively stronger. CRF exhibits larger bias in CR relative to LR with both radiation scheme. In CR$_{CRR}$ and CR$_{SEQ}$, strong negative CRF biases exist over regions of upwelling and the western coast of Australia, whereas strong positive CRF biases occur over the Arctic, the Southern Ocean, the eastern tropical Indian Ocean and the northern equatorial Pacific and Atlantic Ocean. A refinement of horizontal resolution



**a** CR$_{CRR}$ minus GPCP (RMSE= 1.39)   **b** CR$_{SEQ}$ minus GPCP (RMSE= 1.35)

**c** LR$_{CRR}$ minus GPCP (RMSE= 1.24)   **d** LR$_{SEQ}$ minus GPCP (RMSE= 1.17)

mm/day

-6 -5 -4 -3 -2 -1 0 1 2 3 4 5 6

**Figure 18.** Annual bias in total precipitation (mm/day) in the (a, c) concurrent and (b, d) classical radiation experiments relative to the Global Precipitation Climatology Project data set v2.3 (GPCP; Adler et al., 2003) for the period 1980–2013.

largely alleviates negative CRF biases over the upwelling regions, whereas the CRF bias over other regions changes barely. Over the tropics, the concurrent radiation scheme shows a slight increase of the bias compared to the classical radiation scheme.

The sign and spatial structure of CRF bias is largely determined by the SW CRF bias (Figure 20), which are partially compensated by the LW CRF bias (Figure 21). Therefore, increased CRF bias in the concurrent radiation scheme is largely attributed to the response of SW CRF errors (Figure 20c), with negligible contribution from the errors in LW CRF. Across all experiments, there are widespread discrepancies between the spatial distribution of biases in the surface CRF and SAT except over the Tibetan Plateau, where positive CRF bias agrees well with warm the SAT bias. To further explore such inconsistency,

the biases in the net SW radiation at the surface are estimated (Figure 22). SAT biases over the North America, the Tibetan Plateau and central Australia are associated with the excessive net surface SW radiation, resulting from the negative biases of the surface albedo (not shown). In contrast, warm SAT biases over eastern Siberia and tropical South America exist along with deficient SW radiation biases, which suggests a dynamical cause for the biased SAT. A misrepresentation of the atmospheric circulation in ECHAM6.3.05 may be responsible for this discrepancy. The concurrent radiation scheme alleviates the biases



**a** CR$_{CRR}$ minus CERES Surf. (RMSE= 13.65) **b** CR$_{SEQ}$ minus CERES Surf. (RMSE= 13.83)

**c** LR$_{CRR}$ minus CERES Surf. (RMSE= 11.54) **d** LR$_{SEQ}$ minus CERES Surf. (RMSE= 10.87)

-60 -50 -40 -30 -20 -10 0 10 20 30 40 50 60 W/m$^2$

**Figure 19.** Annual bias in total cloud radiative forcing (CRF) in the (a, c) concurrent and (b, d) classical radiation experiments relative to data from Clouds and the Earth's Radiant Energy System Energy Balanced and Filled product (CERES-EBAF) surface fluxes edition 4.0 for the period 2001-2013.

in the net surface SW radiation over land, yet increase the biases over the ocean (Figure 22). Overall, the global RMSE of net surface SW radiation largely increases in CR$_{CRR}$ and LR$_{CRR}$. This implies that a further tuning for the concurrent radiation scheme may be needed.

The zonal mean temperature biases in the troposphere are smaller than in the stratosphere for all experiments (Figure 23). In the lower troposphere (between 850 and 1000 hPa), a cold bias amounting to -4 K occurs over the Antarctic for all experiments.
Thermal biases in the mid troposphere are relatively small. Moderate warm biases up to 2 K extend from the tropics to the Arctic. In the lower stratosphere (between 250 hPa and 100 hPa), all experiments suffer from prominent warm biases over the tropics and mid-latitudes, and severe cold biases over the high-latitudes in both hemispheres. Such biases are significantly reduced in LR (Figure 23c and 23d) relative to CR (Figure 23a and 23b) experiments, consistent with the notion by (Stevens et al., 2013) that zonal mean temperature bias can be significantly reduced by enhancing the horizontal resolution. The concur-





**a** CR$_{CRR}$ minus CERES Surf. (RMSE= 16.83) **b** CR$_{SEQ}$ minus CERES Surf. (RMSE= 17.21)

**c** LR$_{CRR}$ minus CERES Surf. (RMSE= 12.54) **d** LR$_{SEQ}$ minus CERES Surf. (RMSE= 12.04)

-60 -50 -40 -30 -20 -10  0  10  20  30  40  50  60 W/m$^2$

**Figure 20.** As in Figure 19, but for the bias in short-wave radiation fluxes.

rent radiation scheme affects the biases in zonal mean temperature in CR simulations barely, yet slightly reduces the biases in
the lower stratosphere between 40ºS and 90ºS.

The patterns of zonal mean westerly wind biases are common to all configurations (Figure 24) and reflect the meridional
structure of temperature biases (Figure 23). In CR$_{CRR}$ and CR$_{SEQ}$, temperature biases drive a northward shift of westerly
winds between 30ºS and 60ºS. LR$_{CRR}$ and LR$_{SEQ}$ exhibit alleviated westerly wind biases than their CR counterpart due to
reduced temperature bias. In the tropics, westerly wind biases are characterized by overly strong easterly in the low- and mid-
troposphere, and large westerly biases in the upper troposphere. The concurrent radiation scheme intensifies the easterly wind
biases over the tropics in CR and LR (Figure 24a and 24c). As suggested by (Stevens et al., 2013), the tropical bias pattern is
an indication of excessive heating associated with the deep convection.

**a** CR$_{CRR}$ minus CERES Surf. (RMSE= 8.49)   **b** CR$_{SEQ}$ minus CERES Surf. (RMSE= 8.32)

**c** LR$_{CRR}$ minus CERES Surf. (RMSE= 7.66)   **d** LR$_{SEQ}$ minus CERES Surf. (RMSE= 7.52)

-60 -50 -40 -30 -20 -10  0  10 20 30 40 50 60 W/m$^2$

**Figure 21.** As in Figure 19, but for the bias in long-wave radiation fluxes.

## 5.2   Climate Variability

To explore the simulated climate variability by the concurrent radiation, we present an analysis of the ENSO teleconnection and interannual variability of in the extra tropics, in the form of the northern and southern annular modes.

### 5.2.1   ENSO feedbacks and teleconnections

The El Niño/Southern Oscillation (ENSO) is the leading mode of interannual variability in the tropical Pacific. ENSO is mainly characterized by the variations of sea surface temperature (SST) in the eastern and central equatorial Pacific. Multiple nega-
tive and positive coupled atmosphere-ocean processes that either favor or suppress the growth of SST anomalies governs the ENSO behavior (Philander, 1989; Neelin, 1998; Jin et al., 2006). Equatorial Pacific SST anomalies associated with ENSO can affect the tropical convection and results in zonal shifts of the Walker Circulation (Philander, 1989; Bayr et al., 2014). Further, the changes in the convection stimulate Rossby waves that propagates to the mid- and high-latitudes through the atmospheric bridge. Many studies (Guilyardi et al., 2004; Guilyardi, 2006; Guilyardi et al., 2009) have shown that the atmospheric model



**a** CR$_{CRR}$ minus CERES Surf. (RMSE= 18.36) **b** CR$_{SEQ}$ minus CERES Surf. (RMSE= 17.78)

**c** LR$_{CRR}$ minus CERES Surf. (RMSE= 15.05) **d** LR$_{SEQ}$ minus CERES Surf. (RMSE= 12.79)

-60 -50 -40 -30 -20 -10  0  10  20  30  40  50  60 W/m$^2$

**Figure 22.** As in Figure 19, but for the bias in net surface shortwave radiation fluxes.

dominates the simulated ENSO properties in the coupled climate models. Therefore, it is crucial to evaluate the ENSO teleconnections that project the influence of ENSO globally. The covariance between the global DJF (December-January-February) precipitation anomalies and the DJF precipitation anomalies over the Niño 4 region is shown in Figure 25. CR$_{SEQ}$ and LR$_{SEQ}$ typically capture the pattern over the tropical Pacific, tropical Atlantic and North America (Figure 25c and 25e) as indicated by the GPCP observations. The concurrent radiation scheme exhibits similar teleconnection patterns to the classical radiation

scheme, yet influences the magnitude of the response to ENSO. CR$_{CRR}$ underestimates the magnitude of atmospheric teleconnection over the Maritime Continent (Figure 25), whereas LR$_{CRR}$ overestimates the teleconnection over this region (Figure 25). Additionally, CR$_{CRR}$ and LR$_{CRR}$ exhibits artificially positive response in the southern tropical Indian Ocean relative to their counterpart using the classical radiation scheme.

Covariance of DJF geopotential height anomalies at 500 hPa (Z500) and normalized DJF precipitation anomalies over the

Niño 4 region is calculated to investigate the diabatic forcing of the tropical Pacific on the boreal winter atmospheric circulation in the Northern Hemisphere (Figure 26). The ERA-interim depicts positive covariance over Canada and southern Greenland, and negative relations over northeast Pacific, north Atlantic and Siberia. CR$_{SEQ}$ is able to simulate the pattern over northeast



**Figure 23.** Mean bias in zonal-mean temperature in the (a, c) concurrent and (b, d) classical radiation experiments relative to ERA-Interim for the period 1980–2013 (shading) and the climatological mean from ERA-Interim (contours).

**Figure 24.** As in Figure 23, but for the bias in the zonal mean wind.

**a** GPCP Cov. [precip(nino4), precip]

**b** CR$_{CRR}$ Cov. [precip(nino4), precip]   **c** CR$_{SEQ}$ Cov. [precip(nino4), precip]

**d** LR$_{CRR}$ Cov. [precip(nino4), precip]   **e** LR$_{SEQ}$ Cov. [precip(nino4), precip]

-3.2 -2.4 -1.6 -0.8 0 0.8 1.6 2.4 3.2

**Figure 25.** Covariance of DJF (December-January-February) precipitation anomalies with the normalized time series of precipitation anomaly in the Niño4 region for (a) GPCP observations, (b, d) concurrent and (c, e) classical radiation experiments.





Pacific, Canada and Greenland. However, it fails to capture the component over Siberia and the signal over the North Atlantic
is shifted northeastward. Increasing the horizontal resolution to LR shows no improvement for the teleconnection pattern with
the classical radiation scheme. $CR_{CCR}$ retains the main features of ENSO teleconnection as shown by $CR_{SEQ}$, whereas $LR_{CCR}$
exhibits weak response of atmospheric circulation to the diabatic forcing over the tropical Pacific compared to the ERA-interim
(Figure 26d). This implies that a tuning for the convection scheme in the concurrent radiation scheme is required.

### 5.2.2  Northern Annular Mode

The Northern and Southern Hemisphere Annular mode (NAM and SAM, also known as the Arctic and Antarctic Oscilla-
tion) are the leading modes of variability of the extratropical circulation in both hemispheres. Both annular modes explain the
month-to-month and year-to-year (especially the cold-season) variability of the atmospheric circulation, which exhibit pro-
nounced impacts on the climate in the mid-latitudes and the polar region (McAfee and Russell, 2008; Kidston et al., 2009).
The NAM is defined as the leading empirical orthogonal function (EOF) of the DJF sea level pressure (SLP) in the Northern
Hemisphere. As shown by the ERA-interim reanalysis data, the NAM is characterized by zonally symmetric structures (Fig-
ure 27). There are large and positive loadings over the polar cap region, surrounded by zonally ring-shaped negative loadings
centered over the northeast Pacific and the North Atlantic. The leading mode explains 30.4% of total variance. All experiments
show larger explained variance of the NAM than the ERA-interim. Horizontal resolution does not show strong influence on the
NAM pattern, except that $CR_{CRR}$ and $CR_{SEQ}$ exhibit slightly higher explained variance relative to their LR counterparts. The
concurrent radiation scheme captures the centers of action better than the classical scheme, especially over the North Atlantic
sector. The center of action over North Atlantic is shifted eastward in $LR_{SEQ}$ compared to the ERA-interim and $LR_{CRR}$.

The atmospheric teleconnection associated with the NAM is calculated by regressing the DJF SAT anomalies on the time
series of principal components (PC1) corresponding to the leading EOF (Figure 28). Associated with the positive phase of
the NAM, warm SAT anomalies occur over Europe, Siberia and the western United States, while cold temperature anomalies
exist over western Alaska, far eastern Russia, Greenland and eastern Canada (Figure 28a). $CR_{SEQ}$ significantly underestimate
the response over Greenland, Baffin Bay and Siberia, but overestimate the response over western America. Additionally, an
artificially negative response occurs over western Canada. $CR_{CCR}$ share similarities with $CR_{SEQ}$, with minor differences in the
magnitude of the response, which is also seen in the LR experiments. The atmospheric teleconnection simulated by two LR
experiments generally agree better with the ERA-interim than the CR simulations, which reproduces the widespread negative
SAT anomalies over Europe and Siberia (Figure 28d and 28e).

### 5.2.3  Southern Annular Mode

Similarly, the JJA (June-July-August) SAM is defined as the leading EOF of JJA SLP anomalies in the Southern Hemisphere.
Similar to the NAM, the SAM depicts large loadings over the Antarctic and three surrounding centers of action over the
southern Pacific, Indian and Atlantic Ocean at approximately 45ºS (Figure 29a). $CR_{CCR}$ and $CR_{SEQ}$ simulate slightly westward
shifted centers of action over the southern Indian (Figure 29b and 29c). The explained variances by the leading EOFs are





**Figure 26.** As in Figure 25, but for DJF 500 hPa geopotential height anomalies in the Northern Hemisphere.



**Figure 27.** The leading empirical orthogonal functions (EOF) of December-January-February (DJF) sea level pressure (SLP) anomalies calculated from (a) ERA-interim, (b-c) classical and (d-g) asynchronous radiation experiments for the period 1980-2013.





**Figure 28.** Linear regressions of the DJF 2m-air-temperature on the NAM index (normalized PC1) from (a) ERA-interim, (b, d) concurrent and (c, e) classical radiation experiments for the period 1980-2013.





overestimated in these two experiments. A refinement of horizontal resolution improves the simulation of the SAM pattern. Among all experiments, LR$_{\text{CCR}}$ shows the best agreement with the ERA-interim (Figure 29a and 29d). This may be linked to small biases in the mean atmospheric circulation (Figure 24c).

The SAT response to positive phase of SAM is computed by regressing the SAT anomalies on the PC1 of SLP in the Southern Hemisphere (Figure 30). There are warm temperature anomalies over most of the Antarctic and Ross Sea, while cold anomalies exist over the Weddell. LR$_{\text{CCR}}$ and LR$_{\text{SEQ}}$ underestimate the SAT response over the Weddell Sea whereas CR$_{\text{CCR}}$ and CR$_{\text{SEQ}}$ fails to reproduce the SAT response over both the Ross Sea and Weddell Sea (Figure 30b-30e). Overall, an increase in the horizontal resolution improves the simulated climate variability and its atmospheric teleconnections both in the Northern and Southern Hemisphere, however, the concurrent radiation scheme changes these features barely.

# 6  Conclusions

This paper presents the implementation of the concurrent radiation scheme in the atmospheric model ECHAM6 and demonstrates its impact on the performance and stability of the model. A detailed analysis shows that the radiative transfer is a relatively expensive component of the model especially for the CR and LR resolution, which are also used in paleoclimate simulation. Although the component exhibits a higher scalabilty profile, the study reveals that it cannot freely scale to its full potential due to the sequential architecture of the classical ECHAM6. The concurrent radiation scheme, on the other hand, organizes the radiation component in parallel with the rest of the atmospheric physics and dynamics and hides its long computation time. The experiments asserted explicitly a noticeable model speedup across the scaling curve with a strong dependency on the relative computational cost of the radiative transfer to the other atmospheric processes. Unlike the classical scheme, this approach enables the radiation component to adopt any viable domain decomposition arbitrarily which may differ from the main model's configuration. The component can accordingly follow a different scaling scheme and benefit from the higher scalability of radiation calculations. This salient feature can eventually decrease the discrepancy between the radiation time step $\Delta t_{\text{rad}}$ and normal atmospheric time step $\Delta t_{\text{atm}}$ with the objective of creating more physical consistency in the model.

The simulated mean climate and internal climate variability by the classical and concurrent radiation scheme have been evaluated. A suite of AMIP experiments have been performed on CR and LR configurations using the two radiation schemes. In terms of long-term mean state biases, e.g., biases in land surface temperature, precipitation, cloud radiative forcings, and zonal mean temperature and wind, a refinement of horizontal resolution exhibits better agreement with the observations. The concurrent radiation scheme generally yields similar results with the classical radiation, except some minor improvements in the mean atmospheric circulation in the Southern Hemisphere. Regarding to the climate variability and associated atmospheric teleconnections, LR simulations agree better with the ERA-interim than their CR counterparts. The concurrent radiation scheme on LR improves the atmospheric teleconnection to the SAM, which is likely linked to the alleviated bias in the mean circulation. On the other hand, the classical radiation scheme on LR shows better atmospheric teleconnection to ENSO than the concurrent radiation. One possible reason is that the classical radiation scheme in ECHAM6 has been properly tuned towards the observa-



**a** ERA SLP EOF (JJA)    26.1%

**b** CR$_{CRR}$ SLP EOF (JJA)    31.3%

**c** CR$_{SEQ}$ SLP EOF (JJA)    30.5%

**d** LR$_{CRR}$ SLP EOF (JJA)    27.6%

**e** LR$_{SEQ}$ SLP EOF (JJA)    42.0%

hPa/σSLP PC1

**Figure 29.** As in Figure 27, but for the June-July-August (JJA) SLP anomalies in the Southern Hemisphere.





**Figure 30.** As in Figure 28, but for the SAM index (normalized PC1).



tions. To conclude, the concurrent radiation scheme presented in this study substantially improves the scalability of ECHAM6, with the major features in the mean climate and internal variability retained.

*Code and data availability.*

1. The source code of the atmospheric model ECHAM6 adopted to the project PalMod (for the concurrent execution of radiative transfer) and used for generating the plots presented in this paper is available under https://doi.org/10.35089/WDCC/SC_PalMod_ECHAM6 (MPI-M and DKRZ, 2021).

2. The output data for generating the plots presented in this paper is available under https://doi.org/10.5281/zenodo.4589140 (Heidari et al., 2021).

*Author contributions.*

1. MH wrote Abstract, Sections 1, 2, 3, 4 and contributed to Conclusions. He conducted the performance analysis and performed the required AMIP experiments.

2. ZS wrote Section 5 and contributed to Conclusions. He performed the AMIP experiments required in Section 5 and analyzed the output.

3. ED contributed to the AMIP experiment required for the performance analysis.

4. JB wrote the excerpt on the YAXT library and provided detailed advice on the performance analysis.

5. HB had the major role in the funding acquisition and supervision of the project leading to this publication. He also provided detailed advice on the direction of the paper and the performance analysis.

All co-authors contributed to the research and the text in this manuscript.

*Competing interests.* The authors declare no competing interests.

*Acknowledgements.*

1. The authors are grateful to the colleagues at the Max Planck Institute for Meteorology for sharing their results and ideas on implementing a similar approach in the ICON model.

2. The authors are also grateful to Prof. Dr. Thomas Ludwig and his Scientific Computing group at Universität Hamburg for their kind support.

3. The authors would like to express their gratitude to Michael Böttinger for providing the graphics in Figure 1 and 5.

*Financial support.* This research has been funded by the Federal Ministry of Education and Science (BMBF) under grant number FKZ:01LP1515A.



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
