# Peer review of "Concurrent Calculation of Radiative Transfer in the Atmospheric Simulation in ECHAM-6.3.05p2"

_Geoscientific Model Development, 2021_

## Author Response (AR1)

**Authors' response to the comments from referee#1**

We would like to thank the reviewer for the work and his/her comments. We have the following remarks corresponding to the points stated in the interactive comments:

**(1.)**

**Referee#1:** What is the overall strategy to achieve load balancing? Will adapting the spatial and temporal resolution of the radiation calculations always be feasible and sufficient to load-balance with the atmospheric physics? What will be the effect of the suggested "adopting a coarser domain decomposition" (cf. lines 255-258) on the accuracy of the models?

**Authors:** The aim of this manuscript is mainly a report on the implementation of the idea of the concurrent radiation scheme, which offers a solution for improving the performance of the model. Although we generally expect a thorough analysis of this approach in a follow-up paper, there are a few points in this regard that are worth mentioning here. In the concurrent radiation scheme, the radiation component was separated from the main model, and thus enabling the component to opt for a different choice of domain decomposition from the one adopted by the main model. This feature however requires a reordering of data between the MPI processes assigned to the main model and the radiation component which is fully handled by the YAXT library.

Such an approach is a means solely aimed at improving the overall performance of the model and creating a load-balance between the MPI processes assigned to the main model and the component. It is noteworthy that the accuracy of the model will not however be affected if the radiation component and the main model adopt different domain decomposition. This is because the temporal and spatial resolutions of the model and the radiation component are not affected and thus the simulations results are expected to remain bit-wise identical.

Changing the temporal resolution of the radiation component is, on the other hand, motivated by the need for decreasing the gap between $\Delta t_{rad}$ and $\Delta t_{atm}$ in order to achieve a consistency in the model. Since the radiation component scales better than the main model, assigning equal numbers of MPI processes to the calculation of radiative transfer and the rest of the model results in an idle time in the MPI processes assigned to the radiation component and thus creates a load imbalance in the model, as shown in Fig 13. Such an idle time offers an ample opportunity for reducing the radiation time step and calculating radiative transfer more often (without increasing the total execution time of the model or increasing the resource usage). Although this approach is expected to improve the load-balance in the model, it should be noted that this solution is primarily in pursuit of generating a more consistent atmospheric model rather than creating a load-balancing in the model.

In the nutshell, adopting a coarser or finer domain decomposition is the effective means for load-balancing without affecting the accuracy of the model. Changing the temporal resolution of the radiation component is however aimed at improving the accuracy of the model though it can potentially contribute to load-balancing of the model as well.

**(2.)**

**Referee#1:** Related to this, I'm missing a more systematic assessment in section 5 which currently is somewhat qualitative (e.g., "This implies that a further tuning for the concurrent radiation scheme may be needed", lines 321-322: what does this actually mean?)

**Authors:** Model tuning generally refers to the adjustment of a set of model parameters towards the end of model development. It is an approach that is commonly used to obtain certain properties, e.g., temperature, cloud feedback, climate sensitivity, in good agreement with observational records. Without tuning, models may drift away from the realistic state of Earth's climate. As shown by Mauritsen and Roeckner (2020) [1], a set of parameters concerning shallow convection, critical relative humidity in the fractional cloud scheme and mixed-phase clouds are carefully tuned in ECHAM6.3.

Here we adopt the concurrent radiation scheme in the model. which may strongly affect the radiation budget. Yet parameters concerning the cloud scheme are not adjusted. Figure 22 exhibits that the biases in cloud radiative forcing of the concurrent radiation scheme is larger than those of the sequential radiation scheme. This suggests that a specific tuning, especially for the relative humidity threshold for cloud formation in the upper troposphere and the lowest model level (Mauritsen, personal communication), for the concurrent radiation scheme is needed.

We intend to illustrate the concepts of the concurrent radiation scheme and its implementation in ECHAM6.3.05 in this manuscript. A more systematic assessment on the concurrent radiation scheme is somewhat beyond the scope of this study. We would present a thorough assessment in a separate paper together with the tuned model.

**Author's changes in manuscript:**

We reformulate the sentence in lines 321-322. The new lines are in lines 353-356.

"This implies that model parameters concerning the cloud formation should be carefully adjusted for the concurrent radiation scheme. Specifically, the relative humidity threshold for cloud formation in the upper troposphere and the lowest model level should be changed to improve the match with observational records (Mauritsen and Roeckner, 2020)."

**(3.)**

**Referee#1:** How specific is the "tuning" for the specific model setup?

**Authors:** Please refer to our response in (2.). Parameters concerning the relative humidity threshold for cloud formation in the upper troposphere and the lowest model level should be changed.

**(4.)**

**Referee#1:** Will it generalize to other ESM model setups and compute systems?

**Authors:** The idea of concurrent execution of model components has also been evaluated in many other setups including:

- Async output model
- Biogeochemical tracers in the ICON or MPIOM ocean model
- The concurrent calculation of iceberg trajectories in FESOM
- The scheme can also support heterogeneous HPC systems, where some components might run on GPUs and some remain on CPUs.

**(5.)**

**Referee#1:** The new scheme also introduces a change in the operator splitting technique, the potential effects of which are not discussed nor systematically assessed in the paper: In the classical radiation scheme, the radiation timestep takes the most up-to-date quantities of the atmospheric physics as input, whereas in the concurrent radiation scheme the input quantities systematically lag behind by one ATM timestep (cf. Figs 2 and 6)

**Authors:** This is one of the biggest theoretical questions to be answered and we would be very happy to discuss this in a follow-up paper together with climate scientists and mathematicians. However, the current paper should only be seen as a proof of concepts and concentrates on the implementation.

**(6.)**

**Referee#1(minor points):**

- line 143; "receives feedback ... upon the request" what does this mean ?
- line 178: specify type of InfiniBand (EDR, HDR, or alike)
- line 205: "... is adopted to ..." what does this mean?
- typo in line 408: scalability
- Figs 2, 6, 13-16: some of the labelling is hard to read, in particular red font on blue background

**Authors:** All the minor points were applied to the revised version of the manuscript and is ready to be submitted upon the request.

**(7.)**

**Referee#1's minor point:** It was not possible to anonymously (i.e. without registration of an account at DKRZ) download the modified ECHAM6 source code from https://doi.org/10.35089/WDCC/SC_PalMod_ECHAM6 for inspection and for an assessment of reproducibility aspects, but I'd consider this only a minor point, given that ECHAM is such a well-established code in the ESM community.

**Authors:** An account at DKRZ is not needed for downloading the model source code. However, anonymous access to the code is not permitted by the owner (i.e. MPI-M) either. We suggest the following steps:

1. Please kindly accept the license agreement (as also shown in the DOI "use constraints") at https://mpimet.mpg.de/en/science/modeling-with-icon/code-availability.
2. Once MPI-M informs us that the user has accepted the license, we will provide you access to the source code.

**(8.)**

**Reference for reto Referee#1:**

[1] Mauritsen, T., & Roeckner, E. (2020). Tuning the MPI-ESM1.2 global climate model to improve the match with instrumental record warming by lowering its climate sensitivity. Journal of Advances in Modeling Earth Systems, 12, e2019MS002037. https://doi.org/10.1029/2019MS002037.

**Citation**: https://doi.org/10.5194/gmd-2021-117-CC1

We would like to thank the reviewer for the work and his comments. We have the following remarks corresponding to the points stated in the interactive comments:

(1.)

**Referee#2:** Abstract: comment on the potential meteorological impact (even if neutral) due to radiation seeing an older atmospheric state. Currently there is no mention of the impact of the scheme on the scientific content of model simulations.

**Authors:** We insert two sentences addressing the influence of the concurrent radiation scheme on the scientific results in the abstract. Please refer to the following text in Italic.

**Author's changes in manuscript:**

A change starts at line 10:

"…Our experiments show that ECHAM6 can achieve a speedup over 1.9x using the concurrent radiation scheme. *By performing a suite of stand-alone atmospheric experiments, we evaluate the influence of the concurrent radiation scheme on the scientific results. The simulated mean climate and internal climate variability by the concurrent radiation generally agree well with the classical radiation scheme, with minor improvements in the mean atmospheric circulation in the Southern Hemisphere and the atmospheric teleconnection to the Southern Annular Mode.* This empirical study serves as a successful example …"

(2.)

**Referee#2:** L17: Radiation is not always expensive, especially in the case of high resolution weather models: Hogan & Bozzo (JAMES, 2018) reported that 1-hourly radiation accounts for only 3.5% of the computational cost of the ECMWF model at full (9 km) resolution. This is probably largely due to running the radiation on a coarser grid, since the ratio of radiation timestep to model timestep is the same (2h/15min for ECHAM and 1h/7.5min for ECMWF).

**Authors:** We modified L17 to limit the statement to only LR and CR resolutions.

**Author's changes in manuscript:**

The sentence in in L17 is now in line 21:

"Radiative transfer is one of the most expensive parts for coarse and low-resolution atmospheric simulations."

(3.)

**Referee#2:** L21: It is an overstatement to say that the shortwave and longwave are *widely* separated; in fact there is around 12 W m-2 of solar energy at wavelengths longer than 4 microns, which is traditionally regarded as the longwave domain.

**Authors:** L21 was modified to improve the sentence.

**Author's changes in manuscript:**

The sentence in L21 is now in line 25, which is as follows:

"Energy transfer in the atmosphere involves electromagnetic radiation that can be separated into short and long wave parts."

(4.)

**Referee#2:** L35: There are more recent studies than this that might be of interest: on the impacts of coarse temporal sampling in the ECMWF model (e.g. Fig. 6 of Hogan & Bozzo 2018) and how to mitigate them (e.g. Hogan & Bozzo, JAMES 2015; Hogan & Hirahara, GRL 2015).

**Authors:** Thanks to the referee's suggestions, we add a paragraph after L40 (in the old version) to cite the other studies:

**Author's changes in manuscript:**

The added paragraph starts at L45 (in the new version):

"Resolving radiation transfer on coarser time and spatial resolutions can however lead to errors in weather and climate simulations. Authors in (Hogan and Hirahara, 2016) examine the biases that occur due to discrete sampling of solar zenith angle in models which calculate radiation every 3h and propose a careful treatment of the cosine of the solar zenith angle to mitigate the negative impacts. A report by (Hogan and Bozzo, 2015) describes a computationally efficient solution to the problems raised in models that call the radiation scheme infrequently in time or on a reduced spatial grid by updating the surface longwave and shortwave fluxes in every time step and grid point according to the local skin temperature and albedo. A follow-up study (Hogan and Bozzo, 2018) introduces a flexible new radiation scheme (ecRAD) for the ECMWF model which is around 41% faster than the previous package and shows some improvements in the skill of weather forecasts by calling the radiation scheme more frequently for the same overall computational cost."

(5.)

**Referee#2:** L46: Some mention must be made here of the potential down-side of radiation in parallel, which is that the fluxes and heating rates fed to the rest of the model will be "older" by around one radiation timestep than in the traditional approach of radiation in series. The impact on forecast skill was not really studied by Mozdzynski & Morcrette, but could be important. In the ECHAM context, the classical configuration involves radiation fields computed at a particular time being used in the rest of the model for the following 0-2 hours (with some corrections for surface temperature and sun position, but not for clouds). In the concurrent scheme, the radiation fields are not 0-2 hours but 2-4 hours old. The impact on model fields is something you address later in this paper, but it needs to be mentioned here in the introduction as an important consideration. One physical process that benefits from a tighter coupling in time with radiation is boundary-layer clouds, particularly stratocumulus: when they form they are maintained by longwave cooling at cloud top. This could have been one of the reasons why Hogan & Bozzo (2018, Fig. 6) found that calling radiation more frequently led to more skillful forecasts of near-surface temperature *and* low cloud cover.

**Authors:** Thanks to the recommendation by the referee, we improve the paragraph starting at L80 to give the message and augment it by the example and reasoning suggested by the referee.

**Author's changes in manuscript:**

The paragraph in L80 (older version) is now in L91 and contains the change, which is as follows:

"This paper, on the other hand, presents a report on the concurrent radiation scheme applied to the atmospheric model ECHAM6 and provides a thorough analysis on the performance and stability of the model. Calculating radiative transfer in parallel with other atmospheric processes can potentially affect the model's accuracy since the radiation fields will always lag one more radiation time step behind in comparison with the classical scheme. This lag may have negative impacts on physical processes that benefit from a tighter coupling in time with radiation. The boundary-layer clouds, particularly stratocumulus, are a good example. They are maintained by longwave cooling at cloud tops once they are formed. This could explain why (Hogan and Bozzo, 2018) found that calling radiation more frequently led to more skillful forecasts of near-surface temperature and low cloud cover."

(6.)

**Referee#2:** Fig. 1 reproduces Fig. 1 of Giorgetta et al. (2013), except for the addition of a small radiation box - in the interests of shortening the paper it should be removed. Fig. 5 is a small change that doesn't really illustrate the concept of concurrent radiation - all you need is Figs. 2 and 6, which could be combined into a single figure with two panels.

**Authors:** As advised, Fig 1 and 5 were removed in favor of shortening the manuscript. However, combining Fig 2 and 6 revealed little benefit in saving space thus they were left untouched.

(7.)

**Referee#2:** I understand that the red line in Fig. 9 should be the ratio of the red and blue lines in Fig. 8, but it doesn't look like that in that it is always larger than 1.6, when Fig. 8 shows that concurrent radiation is sometimes slower than classical radiation. Is this because the X axis is different, i.e. in one it is the total number of MPI processes and in the other it is the number used for just one part of the model? Surely it should be the total number of MPI processes allocated in both instances, but perhaps I misunderstand something. This needs to be clarified, and a fair comparison shown.

**Authors:** The curves in Fig 9 show the methodical speedup of the model using the concurrent RAD scheme. The methodical speedup means the improved runtime of the model by making use of the concurrent radiation scheme, in contrast to the classical definition of speedup, where additional resources are used for the same computation. The methodical speedup is therefore the ratio of the overall performance of the model using the concurrent radiation scheme divided by the performance of the model using the classical radiation scheme. The X axis shows the number of MPI processes assigned to the concurrent RAD scheme. Half of the resources (shown by X axis) are assigned to the model when it adopts the classical scheme. For each point on the curves, we do the following.

1. The model is configured with the concurrent RAD scheme and allocates a number of resources shown by X-axis. We measure the performance (simulated years per day SYPD) of the model as SYPDconcurrent.
2. Then, the model is configured with the classical RAD scheme and allocates HALF of the number of resources shown by X-axis. We measure the performance (simulated years per day SYPD) of the model as SYPDclassical
3. Methodical speedup = SYPDconcurrent / SYPDclassical

**Author's changes in manuscript:**

We modify the text at L203 (old version), which now becomes L222 in the new version (note that Fig 9 becomes Fig 7 in the new version of the manuscript):

"The red curve in Figure 7 displays the methodical speedup of the model using the concurrent radiation scheme. Here, the methodical speedup means the improved runtime of the model by making use of the proposed concurrency, in contrast to the classical definition of speedup, where additional resources are used for the same computation. The methodical speedup is therefore defined as the ratio of the overall performance of the model using the concurrent radiation scheme (using 2N resources) divided by the performance of the model using the classical radiation scheme (using N resources). On this account, for each point on the speedup curve(s), the number of resources assigned to the model using the classical radiation scheme is half the resources allocated by the model using the concurrent radiation scheme. Hence, the X-axis indicates only the total number of allocated MPI processes to the model if the concurrent radiation scheme is used by the model. However, the model allocates half of the MPI processes shown at the X-axis when it adopts the classical radiation scheme. The red curve shows that…"

(8.)

**Referee#2:** Fig. 16: It is worth pointing out that there would be likely little to be gained in terms of model accuracy from running the radiation scheme in a climate model every 15 minutes. Is this figure needed, since the principle can be explained easily in the text?

**Authors:** We removed Fig 16 as advised and improved the text starting from L255 until the end of section 4 (at L267).

**Author's changes in manuscript:**

The text starting at in L255 (in the old version) now starts at L282 in the new version (note that the figure numbers have already changed in the new text)"

"The concurrent radiation scheme, however, puts forward a general solution to remove the load imbalance between the radiation component and the main model. This solution provides a remedy for the idle time imposed on the main model at some configurations (such as 48, 96, 192, 288 or 384 MPI processes as shown in Figure 10) which exhibit a suboptimal resource efficiency due to the slow calculation of radiative transfer. In this approach, the radiation component is enabled to adopt finer domain decomposition and allocates a higher number of resources (in comparison to the main model) in order to catch up with the fast calculation of other atmospheric processes. By the same token, Figure 13 suggests a configuration in which the radiation component adopts coarser domain decomposition and allocates a lower number of MPI processes compared to the main model. This arrangement is also a remedy to remove the load imbalance at the configurations (such as 576, 768 and 1024 MPI processes as shown in Figure 10) in which the radiation component experiences a long idle time due to the slow calculation of other atmospheric processes.

In addition, the concurrent radiation scheme offers an opportunity for coupling the radiation component to the other atmospheric processes at every normal time step (i.e. $\Delta t$ rad = $\Delta t$ atm ). This feature can ultimately bring the model to the physical consistency between the radiative and physicochemical atmospheric states, albeit probably with a negligible impact on the model's accuracy. It is notable that the current implementation of the concurrent radiation scheme in ECHAM6 already provides the technical support for the adoption of finer or coarser domain decomposition for the radiation calculations. In particular, the YAXT library simplifies the data exchange between the concurrent components with disparate domain decomposition. The scientific viability of these schemes, however, requires further investigations and the results will be presented in a follow-up paper."

(9.)

**Referee#2:** In the evaluation of the concurrent radiation scheme (Figs. 17-30) for a particular variable, the bias is shown for the concurrent and classical model versions, and the reader is expected to try to pick out the differences by eye which is not really possible. Far more useful would be to show the bias for just one of these versions, and then the difference between concurrent and classical, plus, crucially, some stippling to show where the changes are statistically significant. A particular area of interest would be in the marine stratocumulus regions where radiation and cloud processes are coupled on quite a fast timescale.  From what I can see in the figures shown, there appears to be no significant effect of concurrent radiation on any of these variables (except possibly in Fig. 22), but it would really help to show difference plots to be sure.

**Authors:** The differences between the concurrent and the classical radiation is added to the figures as suggested, along with hatching indicating the significance (Figure R1 and R2). The referee is correct that the concurrent radiation does not exhibit much significant effect on the surface temperature or precipitation, nor on the zonal mean temperature and zonal wind.

**Authors' changes in manuscript:**

Please refer to figures 14-19 in the revised manuscript.

[Figure]

**Figure R1.** Annual bias in surface air temperature (SAT) in the (a) concurrent and (c) classical radiation experiments relative to the Research Unit time series 4.01 data set (CRU TS; Harris et al., 2014) for the period 1980–2013. Annual bias in total precipitation (mm/day) in the (b) concurrent and (d) classical radiation experiments relative to the Global Precipitation Climatology Project data set v2.3 (GPCP; Adler et al., 2003) for the period 1980–2013. Differences in (e) SAT and (f) precipitation between the concurrent and classical radiation experiments. Hatching indicates the differences are significant at the 95% confidence interval using Students' t-test.

[Figure]

**Figure R2.** Annual bias in zonal mean temperature in the (a) concurrent and (c) classical radiation experiments relative to the ERA-interim for the period 1980–2013. Contours in (a) and (c) indicate the climatological zonal mean temperature for ERA-interim. Annual bias in zonal mean zonal wind (m/s) in the (b) concurrent and (d) classical radiation experiments relative to ERA-interim for the period 1980–2013. Contours in (b) and (d) indicate the climatological zonal mean zonal wind for ERA-interim. Differences in (e) SAT and (f) precipitation between the concurrent and classical radiation experiments. Hatching indicates the differences are significant at the 95% confidence interval using Students' t-test.

(10.)

**Referee#2:** Figs. 19-21: I don't see the need to show the total cloud radiative effect in addition to the longwave and shortwave components, since the latter two fingerprint specific cloud errors in models, whereas the total is simply a messy mixture of the two. Therefore I suggest Fig. 19 is removed. The captions for Figs. 20 and 21 are misleading as they should say they are the bias in cloud radiative forcing rather than in fluxes.

**Authors:** The figure for total cloud radiative effect is removed. The figure captions for shortwave and longwave radiative forcing are revised as suggested.

**Authors' changes in manuscript:**

Please refer to figure 15 in the revised manuscript. Text from Lines 350-365 are changed accordingly.

(11.)

**Referee#2:** Figs. 19-21 show surface cloud radiative forcing estimated from CERES. The longwave CRF from series is very uncertain at the surface, since there is an assumption about the location of cloud base, which is unknown for passive satellite sensors. It would be much better to compare to the top-of-atmosphere CRF values from CERES, which are much closer to what the satellite measures. Note that in the shortwave, the surface and top-of-atmosphere CRF values are very similar.

**Authors:** We thank the referee for this comment. We now replace the surface CRF variables with those at the top-of-atmosphere from CERES.

**Authors' changes in manuscript:**

Please refer to figure 15 in the revised manuscript.

(12.)

**Referee#2:** In the interests of reducing the number of figures, is there really a need to show both the CR and LR models? While the difference between these two resolutions is interesting to some, it is not the topic of this paper and is a bit of a distraction from the effect of concurrent radiation. As far as I can tell, concurrent radiation has only a limited effect at either resolution, so isn't it enough to show just one resolution and then say that concurrent radiation doesn't affect the other one much either? Or possibly plots for one of the resolutions could be consigned to Supplementary Material?

**Authors:** We agree that the concurrent radiation exhibits very limited influence on the LR and CR configuration as shown in the AMIP experiments. We now keep the figures for the CR configuration in the manuscript, yet move the figures for the LR configuration to the supplement as suggested.

**Authors' changes in manuscript:**

Please refer to figures in the manuscript for the CR configuration and figures in the supplement for the LR configuration.

(13.)

**Referee#2:** Why do the plots for the LR model, which should be higher resolution than CR, all show Gibbs fringes? Is this a genuine feature of the model fields, or is it some kind of degradation in the extraction or plotting?

**Authors:** It is a genuine feature for ECHAM6 at LR configuration. Please refer to Figure 5 in Stevens et al. (2013). This is a longstanding problem in ECHAM, which is linked to the poor representation of cloud formation in the major stratocumulus region.

(14.)

**Referee#2:** Figs. 26-30: Are all of these figures needed? I would have thought that in the interests of shortening the paper some could be omitted, especially if there is no significant effect of concurrent radiation, in which case the results could simply be stated in the text.

**Authors:** Figures 26-30 exhibits the simulated seasonal and interannual variability by the concurrent and classical radiation scheme. This could be of interest to some readers. Yet we understand the necessity to shorten the manuscript. Therefore, figures for the LR configuration are moved to the supplement. We then produce three new figures to replace Figures 25-30. We hope this could satisfy the referee.

**Authors' changes in manuscript:**

Please refer to figures 18-19 in the revised manuscript and figures S5-S6 in the supplement.

**Reference:**

Stevens, B., et al. (2013), Atmospheric component of the MPI-M Earth System Model: ECHAM6, J. Adv. Model. Earth Syst., 5, 146– 172, doi:10.1002/jame.20015.